# Troponin Test, Not Only a Number: An Unusual Case of False Positive

**DOI:** 10.3390/ijms252211937

**Published:** 2024-11-06

**Authors:** Michela Salvatici, Monica Gaimarri, Francesca Rispoli, Barbara Bianchi, Delia Francesca Sansico, Eleonora Matteucci, Andrea Antonelli, Francesco Bandera, Lorenzo Drago

**Affiliations:** 1UOC Laboratory of Clinical Medicine with Specialized Areas, IRCCS MultiMedica Hospital, 20138 Milan, Italy; michela.salvatici@multimedica.it (M.S.); monica.gaimarri@multimedica.it (M.G.); francesca.rispoli@multimedica.it (F.R.); barbara.bianchi@multimedica.it (B.B.); deliafrancesca.sansico@multimedica.it (D.F.S.); eleonora.matteucci@multimedica.it (E.M.); 2Coronary Unit, IRCCS MultiMedica, 20099 Milan, Italy; andrea.antonelli@multimedica.it (A.A.); francesco.bandera@unimi.it (F.B.); 3Department of Biomedical Sciences for Health, University of Milan, 20133 Milan, Italy; 4Clinical Microbiology and Microbiome Laboratory, Department of Biomedical Sciences for Health, University of Milan, 20133 Milan, Italy

**Keywords:** troponin, false positive, heterophile antibodies, cardiovascular risk

## Abstract

Heterophile antibodies, which can arise from infections, autoimmune disorders, or exposure to animal antigens, can interfere with immunoassays. These antibodies can cross-react with the test reagents used in troponin assays, causing a false elevation in troponin levels. The paper describes a case of a 37-year-old male drug abuser admitted to the emergency room with chest pain. A series of troponin measurements performed using different assays gave discrepant results. Only thanks to the use of Scantibodies HBT tubes, which remove heterophile antibodies, was it possible to make a correct diagnosis of troponin negativity. In conclusion, a correct laboratory/clinical approach to the identification of heterophile antibody interference is essential for accurate troponin testing in order to avoid false positive results. Implementing neutralizing tests can significantly improve the reliability of these diagnostic assays, ensuring better patient outcome.

## 1. Introduction

Troponins are a component of the contractile apparatus in both skeletal and cardiac myocytes that regulates and facilitates the interaction between actin and myosin filaments. In the late 20th century, researchers recognized that cardiac-specific isoforms of TnI and TnT could be detected in the blood following myocardial injury [1,2,3]. This discovery was ground-breaking, providing a more sensitive and specific marker for myocardial infarction than previously used biomarkers, such as creatine kinase-MB (CK-MB). The clinical significance of troponin became apparent, leading to its adoption in guidelines for diagnosing MI and changing, over the years, our view of the pathophysiology of myocardial infarction (MI), the diagnostic algorithm of the differential diagnosis of acute myocardial infarction (AMI), and also the monitoring of patients with acute coronary syndromes (ACS) [4,5,6,7,8,9,10]. Finally, the recent development of high-sensitivity assays for cardiac troponin in point-of-care testing (POCT) methods will provide almost immediate results, which is critical in acute settings where time is a crucial factor in patient outcomes [11]. POCT devices are portable and easy to use and the early detection of elevated troponin levels can lead to quicker initiation of appropriate treatments, such as reperfusion therapy in MI patients. Despite its significant impact, the use of troponin as a biomarker is not without challenges. Elevated troponin can also be found in other pathological conditions, such as myocarditis, heart failure, pulmonary embolism, cardiotoxicity by chemotherapy treatment, and arrhythmias, but also in a non-cardiac context, including renal failure and sepsis, as well as unpredictable analytical interferences [12,13,14,15,16,17,18,19,20,21,22,23,24,25,26,27,28,29,30,31,32,33,34,35,36,37,38,39,40,41,42,43,44,45,46]. Fibrin clots, the presence of hemolysis, lipemia, elevated alkaline phosphatase activity, bilirubin, rheumatoid factor, heterophile and human anti-mouse immunoglobulin antibodies (HAMA), and the formation of macro immune complexes, although rare conditions, may have an interfering effect on immunometric assays, regardless of manufacturer, resulting in false positive or negative results with potential misdiagnosis, unnecessary treatments or diagnostic procedures, and increased healthcare costs [19,20,21,22,23,24,25,26,27,28,29,30,31,32,33,34,35,36,37,38,39,40,41,42,43,44,45,46]. Antibody interference in immunoassays is a well-known phenomenon that can affect all analytes and assays. The key to detecting this interference is clinical observation, specifically, when the test result does not align with the clinical picture. In this paper, we present the clinical case of a 37-year-old male user of drugs of abuse, admitted to the emergency room for chest pain with persistently elevated TnI values. The case illustrates a false positive troponin I result, caused by analytical interference, showing how a critical approach in the interpretation of an analytical measurement can be of fundamental support for correct clinical management of the patient.

## 2. Clinical Case

A 37-year-old man with obesity, anxious-depressive syndrome, a history of active drug abuse (cocaine), and a smoking habit, presented to the Emergency Department of IRCCS MultiMedica Sesto San Giovanni Hospital, with chest pain irradiated posteriorly and to the left arm from 1 h. ECG excluded significant ST-segment changes suggestive of acute coronary syndrome and a CT scan was negative for pulmonary embolism, aortic syndrome, and coronary artery disease.

At admission, laboratory testing showed an elevated white blood cell count of 17.05 × 10^9^/L, hemoglobin of 17.2 g/dL, a normal platelets count of 343 × 10^9^/L, an elevated C-reactive protein level of 2.3 mg/dL, an international normalized ratio of 0.99, and normal electrolytes. The chest pain led to TnI to be assayed with a point-of-care testing analyzer (Stratus^®^ CS, Siemens Healthcare Diagnostics), which disclosed a marked elevated TnI value (1.79 ng/mL; reference limit (RL) < 0.04 ng/mL), confirmed in a new sample collected 3 h later (TnI 1.78 ng/mL). At a subsequent monitoring (six hours after the first sampling) the emergency physician ordered the collection of a new sample of the patient for assessing a complete panel of biomarkers at the routine MultiMedica laboratory. The cardiac biomarkers measured at the routine laboratory, on a Siemens Healthineers Atellica CI assay, gave the following results: high sensitive troponin I (hs-TnI) negative (2.6 ng/L; RL 2.5–53.53 ng/L), creatinkinase 475 U/L (RL 15–171 U/L), CK-MB negative (3.50 ng/mL; RL 0.18–5.00 ng/mL), and CK MB Ratio: 0.74% (RL 0.1–4.00%). Due to the discordant results and the clinical presentation, the patient was admitted to ICU for monitoring.

The TnI measurement was repeated on an additional sample (sample 4) both at the routine laboratory and at the POCT laboratory, confirming the discordant result between TnI values obtained on the Atellica IM and the Stratus^®^ CS (2.6 ng/L; RL 2.5–53.53 ng/L, vs. 1.73 ng/mL; RL < 0.04 ng/mL, respectively). Therefore, the TnI values obtained on the Stratus^®^ CS, were permanently markedly elevated, although without displaying a typical rise and fall of troponin plasma levels, while TnI values obtained on the Atellica IM were permanently negative also in the following two days of evaluating (Table 1).

Despite the negative result of hs-TnI, with the suspicion of acute myocardial damage due to myocarditis or pericarditis, the subject underwent a cardiac MRI. No signs of acute myocardial edema and late gadolinium enhancement were found. Clinically, the patient improved without recurrence of chest pain and there were no further ECG changes or abnormal echocardiographic findings.

Based on this clinical and laboratory picture, as raised TnI results did not fit the patient’s clinical conditions, the presence of a false positive result was suspected. Cardiologists and laboratory staff in collaboration decided to carry out further investigations on this unusual finding. To check the sample for possible pre-analytical interferences (such as the presence of fibrin, hemolysis, lipemia, and elevated bilirubin), as a first step, sending samples for analysis on a different immunoassay testing platform was considered to objectively demonstrate an analytical interference. The 4th patient sample was sent to an external collaborative institution to test the concentration of troponin with hs-TnI Alinity ci (Abbott Laboratories, IL, USA) assay. The sample, which had provided a strongly positive concentration of TnI on Stratus CS, gave a negative result of 1.7 pg/mL (RL < 34.2 pg/mL) on hs-TnI Alinity ci, supporting the suspicion of the presence of an interfering substance.

Therefore, this sample was successively treated with specific heterophilic antibody-blocking tubes (HBT, Scantibodies Laboratory Inc., Santee, CA). In brief, 500 μL of patient plasma was pipetted into a specific tube containing the blocking reagent, the tube was gently inverted five times, incubated for 1 h at 18–28 °C, and then tested using Stratus CS. The procedure showed a marked decrease in TnI values from untreated to HBT-treated samples (1.73 ng/mL versus 0.18 ng/mL, respectively) confirming the presence of interference by heterophile antibodies. Finally, the residual plasma was frozen and stored, which allowed us to carry out further investigations on another POC assay, the hs-cTnI Atellica^®^ VTLi hs-cTnI from Siemens Healthineers, Erlangen, Germany. The troponin result with this method was also negative at 6.8 ng/L (RL < 27.1 ng/L) (Figure 1).

## 3. Discussion

In this study, we reported an interesting case of false-positive troponin value due to heterophile antibodies in a 37-year-old patient who arrived at our hospital’s Emergency Department with chest pain. The case report underlines a critical issue, although rare, that should be considered when using immunometric assays: the possibility of interference factors, such as by heterophile antibodies [19,20,21,22,23]. Troponin I (TnI) and Troponin T (TnT) are widely regarded as highly sensitive and specific markers of myocardial damage. However, when diagnostic tests yield inconsistent or inconclusive results—meaning the laboratory findings do not align with the clinical presentation—and a non-dynamic pattern of troponin is observed, it raises suspicion of a false troponin result caused by analytical interference. These analytic interferences include fibrin clots, microparticles in the sample, heterophile and human anti-animal antibodies, rheumatoid factor, interference by bilirubin, hemolysis, lipemia, elevated alkaline phosphatase activity, and macro immunocomplex formation [19,20,21,22,23,24,25,26,27,28,29,30,31,32,33,34]. Understanding the mechanisms behind false troponin measurements, as well as knowing how to identify, confirm, and mitigate these interferences, is crucial in clinical practice. An extensive review by Lippi et al. [36] examined 16 studies and clinical cases, focusing on the impact of heterophile antibodies on troponin levels. These antibodies have multispecific activity and are produced against poorly defined antigens, binding nonspecifically to assay antibodies, resulting in false-positive readings in troponin assays. Heterophile antibodies may arise due to exposure to various antigens such as transfused blood, vaccines, exposure to animals (e.g., mice and rabbits), certain diets, medications, viral infections, rheumatoid factors, autoimmune diseases, and dialysis [36,37,38,39]. The exact prevalence of heterophile antibodies remains unclear, though their interference with troponin assays has traditionally been considered rare. Some studies estimate that false-positive troponin results due to heterophile antibodies could occur in as many as 3.1% of routine tests [33]. Researchers also suggest that this prevalence may rise in the future due to the growing use of immunotherapy and diagnostic techniques involving antibodies. The effect of heterophile antibodies is unpredictable and can influence both Troponin I (TnI) and Troponin T (TnT) testing systems from any manufacturer, as it is related to specific aspects of a manufacturer’s assay, such as the affinity and specificity of the antibodies, the number and sequence of washes, the detection method, and the type of tracer used [36,37,38,39,40,41,42,43,44,45,46]. Immunoassays typically employ two-site reactions (sandwich) that utilize two specific capture and label antibodies targeting the analyte of interest. The capture antibody attaches to any cardiac troponin present in the sample, after which the label antibody is introduced and binds to the captured cardiac troponin, producing a detectable signal that is used to ascertain the concentration of cardiac troponin. Heterophile antibodies can form complexes with both the capture and label antibodies of the analyte, effectively linking them together and resulting in a false outcome. The most effective method to detect false-positive troponin results caused by heterophile antibodies is by treating the sample with heterophile antibody blockers (HBT). Lum et al. [40] reported the case of a 57-year-old man with persistently elevated troponin I levels that were not consistent with clinical findings of myocardial injury. After treating the blood sample with HBT, the troponin level dropped significantly. Similarly, Bionda et al. [41] reported a case of a 51-year-old man with an ECG showing no ischemic damage but elevated TnI levels that decreased after treatment with HBT. However, a simpler initial approach could be to rerun the test on a different analyzer or dilute the sample using a zero calibrator or a negative troponin patient sample to check for linearity. If a different analyzer yields different results, or if the diluted sample’s troponin level does not decrease linearly as expected, this suggests that the result was falsely elevated due to antibody interference. In a more recent case, Lakusic et al. [42] presented a 53-year-old woman with elevated troponin I levels despite no clinical evidence of ischemic heart disease. A plateau of troponin levels without the typical dynamic rise and fall led to the suspicion of heterophile antibody interference, which was confirmed when troponin T levels, measured using a different method, were found to be normal. Our case was similar in that TnI levels were elevated on one assay but normal on others. We analyzed the same sample using two different high-sensitivity troponin I methods (Atellica CI and Abbott Alinity ci) and performed a comparison with the Atellica VTLi method, designed for point-of-care testing. We also pre-treated the sample with HBT, which reduced the troponin I result by 90%, indicating successful blockage of heterophile antibody interference. These cases demonstrate the importance of proper management of laboratory data and communication between clinicians and laboratory staff. Laboratories should be aware of the limitations of the assays they use and be prepared to investigate discrepancies by comparing different methods or applying additional treatments when necessary. Adopting a critical approach to routine testing can ensure more accurate diagnoses and improve patient outcomes. False-positive troponin elevations can have significant clinical implications, impacting both patient management and healthcare resources. Understanding these implications is crucial for clinicians to minimize unnecessary interventions and optimize patient care. False-positive results may lead to misdiagnosis prompting unnecessary treatments, exposing patients to risks associated with unnecessary interventions, extended hospital stays, and causing additional consultations that increase healthcare costs. Finally implementing a multiple biomarker system to detect myocardial injury could present a promising approach to enhance diagnostic accuracy and patient outcomes. By integrating various biomarkers, we can capture a broader spectrum of pathophysiological processes involved in myocardial injury, leading to more precise identification of cardiac events. This strategy may help overcome limitations associated with individual biomarkers, such as sensitivity and specificity issues, and can improve the differentiation between cardiac and non-cardiac causes of elevated troponin levels [47,48].

## 4. Conclusive Remarks

The introduction of troponin in laboratory diagnostics has changed the cardiovascular medicine approach. From its discovery as a biomarker of MI to the development of high-sensitivity assays, troponin has become an indispensable tool in the diagnosis, management, and prognosis of heart disease. However, although it is a test with consolidated clinical and analytical value, as with other immunoassays, it can be affected by interferences, leading to spurious cases of false-positive troponin, which clinicians must consider. Therefore, when diagnostic investigations are inconclusive or discrepant with the laboratory results, a close collaboration between clinicians and laboratory staff is of primary importance to avoid harmful investigation and unnecessary treatment for patients.

## Figures and Tables

**Figure 1 ijms-25-11937-f001:**
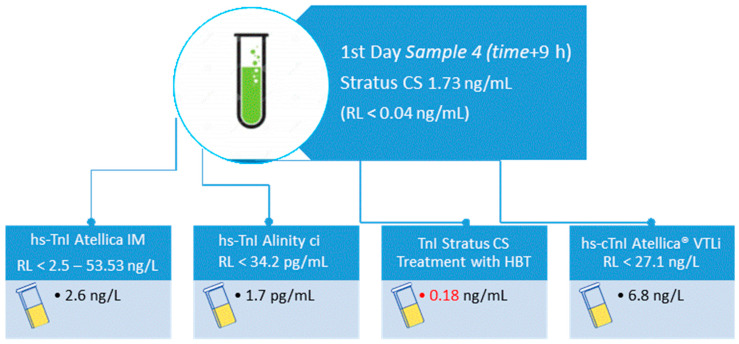
Approach followed by the laboratory to investigate the possibility of a false positive troponin value. 1. Re-assay the sample with another manufacturer’s assay system 2. Submit specimen to another laboratory that employs a different assay for troponin 3. Pre-treat sample with specific heterophilic antibody (HTB)-blocking tube before retesting 4. Re-assay the sample with another POC assay system.

**Table 1 ijms-25-11937-t001:** TnI measurements: time of sampling and results obtained during hospital admission.

	TnI Stratus CS RL < 0.04 ng/mL	Hs-TnI Atellica IM RL < 2.5–53.53 ng/L
**1st Day**		
**Sample 1, Time 0**	1.79 ng/mL	
**Sample 2 Time: +3 h**	1.78 ng/mL	
**Sample 3 Time: +6 h**		2.6 ng/L
**Sample 4 Time: +9 h**	1.73 ng/mL	2.6 ng/L
**2nd Day**		2.6 ng/L
**3rd Day**		<2.5 ng/L

## Data Availability

Data are available inside the manuscript.

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
