# Peer review of "Troponin Test, Not Only a Number: An Unusual Case of False Positive"

_ijms, 2024, doi:10.3390/ijms252211937_

Round 1
Reviewer 1 Report
Comments and Suggestions for Authors
In the case report, the authors reported the value of false positive results of TrI evaluation. The case is undoubtedly impressive and cover important aspect of the biomarker-based approach to myocardial injury detection. However, it remaines unclear the following:
1. The authors might discuss other plausible causes of false positive troponin test, such as fibrin clots, heterophile antibodies, alkaline phosphatase, rheumatoid factor, but not only cross-reactions of diagnostic antibodies.
2. Please, add your opinio regarding the implementation of multiple biomarker system to detect myocardial injury
Reviewer 2 Report
Comments and Suggestions for Authors
In this paper, the authors descrbe a case of a false positive result concerning Troponin measurement using the point of care (POC) Stratus from Siemens, contrasting with normal Troponin levels using the method on classical analyzers (Atellica from Siemens). Interestingly they observed normal Troponin level with the POC of Abbott, and they observed that the high troponin levels of Siemens POC were decreased when patient was collected onspecific tubes that remove heterophile antibodies. Additionnaly, clinical and cardiologic evaluations ruled out a myocardial infarction.
This is an interesting study, I have only 2 minor comments;
In the discussion, line 143, the authors write that "some studies estimate that false-positive troponin results due to heterophilic antibodies could occur in as many as 3.1% of routine tests". References would be welcome.
- Line 148, "Lum et al[40] a 57-year-old man....a verb is most certainly missing
Reviewer 3 Report
Comments and Suggestions for Authors
The involvement of heterophile antibodies in causing false positives in troponin assays is well-documented in the literature. The use of Scantibodies HBT tubes, which effectively remove heterophile antibodies, has also been previously reported. Therefore, the manuscript lacks novelty.
Round 2
Reviewer 3 Report
Comments and Suggestions for Authors
The manuscript is well-written but addressing following comments can improve it further.
1. Why are the different analyzers giving different results? Is the discrepancy associated with the analyzer or sample? Were the different analyzers using different principles of measuring the troponin? Discuss all these aspects in detail in methodology and discussion sections.
2. Provide a rationale for re-assaying troponin with multiple platforms, include a brief explanation on why each specific assay platform was chosen and their known susceptibility to interference. Provide sufficient details regarding each platform and assay’s analytical sensitivity, specificity, and performance characteristics such as coefficient of variation and cross-reactivity rates.
3. Why only HBT approach was chosen as a method for interference-reduction, why not other potential reasons such as fibrin clots, presence of hemolysis, lipemia, elevated alkaline phosphatase activity, bilirubin, rheumatoid factor etc., were considered.
4. Were any control samples (e.g., patient samples with no interference) included during the study to show that this approach was specific to heterophile antibody interference.
5. Is the Alinity ci assay known to have low susceptibility to heterophile antibodies, if yes, why this method is not routinely used for analysis of clinical samples?
6. Discuss the clinical implications of false-positive troponin elevations and provide guidance for the clinicians about the better practice of troponin test. Discuss: is it important to always use HTB blocking tubes in the analysis? How to minimize the false positives and false negatives of troponin test?
